# Mining and quantitative evaluation of the laboratory biosafety policy in China

**Cheng Xiang, Qing Zhu, Jinqing Wen, Lixiang Xie, Rongrong Hao, Xueting Qiu\*, Fei Zhu** \*

Zhejiang Provincial Center for Medical Science Technology & Education Development, Hangzhou, Zhejiang, China

\* hz_qxt@sina.com (XQ), kjzxzf@163.com (FZ)

## Abstract

Policies play a pivotal role in guiding and overseeing laboratory biosafety management. To ensure that laboratory biosafety management is underpinned by scientifically robust and well-founded policies, an analysis and evaluation of existing policies were conducted. The object was to identify their merits and limitations, thereby offering references for future policy development. The qualitative and quantitative analysis were employed to explore 137 central-level policies issued in China as of April 30, 2024. Additionally, based on policy evaluation theory, a PMC index model was established to evaluate 11 representative laboratory biosafety policies. The results showed that: Firstly, these policies, promulgated by 24 distinct departments, spanned three regulatory tiers: laws, regulations, and administrative rules. Secondly, content analysis revealed three primary aspects: (1) management systems, (2) facility, equipment and containment barrier, and (3) operational technical standards. Thirdly, the average PMC index of the 11 policies was 5.05. Specifically, two policies were deemed excellent, eight policies were acceptable and one was inadequate. The low score was mainly attributed to three indicators: policy level, policy timeliness, and policy content. To sum up, laboratory biosafety policies in China were generally rational and comprehensive. However, insufficient collaboration among departments during policy formulation, as well as the need to improve policy continuity were identified. To enhance biosafety laboratory management, four recommendations are proposed: 1. Strengthen communication among different departments; 2.Optimize the policy formulation process; 3. Enhance supervision of biosafety level 1 and 2 (BSL-1/2) laboratories; 4. Harnessing the power of industry associations.

## Introduction

In recent years, driven by the continuous development of scientific research and the recurrent emergence of novel infectious diseases, the number of biosafety

**Data availability statement:** All relevant data are within the paper and its Supporting information files.

**Funding:** The author(s) received no specific funding for this work.

**Competing interests:** The authors have declared that no competing interests exist.

laboratories has exhibited a rapid upward trajectory globally [1,2]. Concurrently, this growth had been accompanied by an escalating number of biosafety incidents, which not only posed a direct threat to the lives and well-being of laboratory personnel but also presented significant challenges to public health [3]. Notable examples underscored this concern. In 2003, a severe acute respiratory syndrome (SARS) infection occurred in a laboratory at the National University of Singapore [4]. Subsequently, in 2005, four staff at Mudanjiang Medical College were infected with hemorrhagic fever [5]. In 2007, another incident of hemorrhagic fever with renal syndrome (HFRS) laboratory infection transpired at a university laboratory animal center in Guangzhou [4]. In 2010, during a "sheep vivisection anatomy experiment" conducted by 30 students at the College of Veterinary Medicine, Northeast Agricultural University, 28 of them were infected with Brucella [6]. These incidents reminded us that effective laboratory biosafety management was imperative.

Policies are essential for guiding and regulating laboratory biosafety practices [7]. They offered an institutional framework for laboratory biosafety management by delineating legal boundaries, formulating operational protocols, implementing oversight mechanisms, and providing support and incentive measures. Numerous countries and organizations have made efforts to enhance laboratory biosafety management. In 1974, the Centers for Disease Control and Prevention (CDC) and National Institutes of Health (NIH) jointly issued *Classification of Etiologic Agents on the Basis of Hazard*. This document categorized pathogenic microorganisms for human research and experimental activities into four categories for the first time [8]. In 1977, the Medical Research Council (MRC) of Canada published the *Laboratory Biosafety Guidelines*, offering policy and regulatory guidance to affiliated laboratories [9]. In 1983, the World Health Organization (*WHO)* released the first edition of the *Laboratory Biosafety Manual (LBM),* which significantly propelled the development of laboratory biosafety globally [10]. The following year witnessed the release of *Biosafety in Microbiological and Biomedical Laboratories (BMBL)* by CDC. *LBM* and *BMBL* have become the basic references for biosafety laboratory construction, and are widely adopted by countries and regions with limited experience in establishing and managing biosafety laboratories [11,12]. In 1995, the Advisory Committee on Dangerous Pathogens (ACDP) in the United Kingdom issued the *Categorisation of Biological Agents*, providing guidance for constructing and regulating Biosafety Level 4 (BSL-4) laboratories. As for China, laboratory biosafety management had a relatively recent origin. The first standard for biosafety laboratory management was introduced in 2002 [13]. After the outbreak of SARS in 2003, China recognized the importance of laboratory biosafety and subsequently promulgated a series of policies. These policies aimed to promote the construction of biosafety laboratories, standardize laboratory certification and accreditation, and enhance regulatory oversight [14]. Under the support of these policies, significant progress has been achieved. Research indicates that China's laboratory biosafety management has entered a standardized stage [15]. Another study indicated that a management system centered on security construction was initially formed [16]. Additionally, investigations reveal effective utilization and maintenance of laboratory facilities and equipment,

along with proper waste disposal practices [17,18]. Despite these achievements, challenges and deficiencies persist in China's laboratory biosafety management. Liu highlighted issues such as an incomplete management system, insufficient awareness and training among laboratory's personnel, and the continued reliance on imported key protective equipment [19]. In this context, numerous studies have reviewed the evolution of China's laboratory biosafety management system and conducted in-depth analysis for specific policies. Up to now, existing researches on laboratory biosafety management policies can be classified into two distinct categories.

The first category of existing research focused on the analysis of policy texts, which could be further divided into two sub-streams: studies on the policy development system within the field of laboratory biosafety, and analyses of specific laboratory biosafety policies. In terms of the policy development system,  it was determined by aggregating and reviewing national regulations. For instance, Zhao conducted a systematic review of the development of the China's high-level biosafety laboratory management systems. The study identified several critical issues within the existing policies, including limited enforceability of guidelines in legal and regulatory frameworks, inconsistencies in pathogenic microorganism classification with internationally accepted protocols, and a lack of standardized operational benchmarks [20]. By reviewing the policies of western countries, such as those of the United States and the United Kingdom, these researchers proposed directions for the future development of China's laboratory biosafety management system. Other researchers conducted cross-country comparative analysis to pinpoint existing challenges. Zhang identified the disparity between China's biosafety laboratory management standards and those of developed countries [16]. Liu pointed out issues such as suboptimal system management, incomplete components, and non-standardized operations. Cao combed through the standards of biosafety laboratories and noted that the standard system remained incomplete, lacking a top-level design and overarching development plan [21]. As for the analysis of specific policies, there were also notable studies. Hu employed a SWOT (strengths, weaknesses, opportunities and challenges) framework to evaluate the filing policy, identifying critical limitations such as inadequate dissemination, high implementation costs, and underutilization [22]. Drawing on the current status of approval and management of highly pathogenic microorganism, Chen conducted a comprehensive assessment of the approval and management system for experimental activities involving highly pathogenic microorganisms. This evaluation clarified the mechanism underpinning the pathogenic microbial management system and identified that deficiencies persisted in both the approval and management approaches for highly pathogenic microorganisms [23].

The second category encompassed policy analyses grounded in the current status of laboratory biosafety management and centered on addressing identified operational challenges. For example, Huang employed a literature-based method to examine the construction and management of mobile biosafety laboratories. This study identified gaps in China's regulatory standards for mobile biosafety laboratories, emphasizing the need for enhanced design, maintenance, and operational protocols [24]. A study on the inspection of pathogenic microorganisms within Hunan Province's health system revealed systemic deficiencies, including lax management of bacteria (virus) strains and infectious materials, non-compliant waste disposal practice, incomplete qualifications among experimental personnel, and insufficient personal protection measures, which were partly attributed to the absence of a robust biosafety management system [25]. Moreover, Liu analyzed the prevailing scenario of biological risk assessment and management in laboratories. The analysis emphasized the imperative need to establish an international framework for biological safety risk management, which would enable comprehensive evaluation and supervision of Biosafety Level 4 (BSL-4) laboratories [26].

Overall, existing studies have extensively explored laboratory biosafety management policies. However, these analyses were based on a small number of policy samples. Additionally, there was a scarcity of research evaluating policies from a textual perspective. This study aims to conduct a comprehensive analysis and evaluation of the laboratory biosafety management policies based on quantitative and qualitative methods. The contribution of this paper is two-fold. Firstly, the quantitative and qualitative analysis based on a great number of policies is more helpful to understand the connotation and objectives of the policy, thereby complementing previous studies with smaller sample sizes. Secondly, we developed

                                                    

a PMC-index model for evaluation of laboratory biosafety management to identify the advantages and disadvantages of laboratory biosafety policies, providing actionable insights for policy improvement and refinement.

## Study design

### Data acquisition and preparation

Since national policies have a wider influence and local policies are formulated under their guidance, our analysis was confined to national policies. We primarily collected data from two professional databases--Peking University Database and Wolters Kluwer, with "*pathogenic microorganism laboratory*", "*laboratory biosafety*" as search terms. Besides, the policies mentioned in academic literature, published on official government websites, and available in public records were incorporated as supplementary sources. Considering that the management of biosafety laboratories involved a numerous standards, mandatory and comprehensive national standards were also included. The policies issued prior to April 30, 2024 were included. A total of 429 policies were ultimately obtained.

To guarantee the authority, representativeness and relevance of the policies included in the study, a systematic exclusion process was implemented. Firstly, we reviewed the full text of the policies. Policies with titles suggesting relevance but whose content deviated from the research topic were excluded. Secondly, for policies with updated versions, only the latest iteration was retained. Additionally, to ensure the integrity and authority of the analyzed policies, informal policies such as discussion drafts, commentaries, official replies, letters and requests, were also removed. Through this multi – stage screening process, a total of 137 policies were selected for subsequent analysis.

### Statistical analysis

Content analysis and PMC index model were employed for evaluation of the laboratory biosafety management policies. A detailed description was provided below.

### Content analysis

Content analysis is a method describing content objectively, systematically, and quantitatively. It can transform qualitative textual information into quantifiable data [27]. It mainly involves identifying the policy texts of the study subject in a specific field, focusing on their content features and core elements, and thus capturing the focus of policy implementation. Presently, content analysis has been widely applied in textual analysis of public policies across diverse sectors, including energy and healthcare. In this study, content analysis was used to elucidate the overarching characteristics of laboratory biosafety-related policy documents in China, with the help of Rost CM6 software.

Semantic social network, rooted in graph theory concepts and methodologies, abstracts entities and the relationships into graph structures. It enables researchers to identify the frequency of key concepts within texts and their contextual associations, thereby facilitating a deeper understanding of the policy framework [28]. In this network, each node represents a keyword, while connecting lines denote co-occurrence relationships between keywords. The size of the nodes and the presence of edges reflect keywords prominence and inter-conceptual linkages. Rost CM6 was employed to generate a co-occurrence matrix, and NetDraw was utilized to visualize the semantic social network, thereby mapping the thematic attributes of laboratory biosafety management policies.

### PMC index model

Policy Modeling Consistency (PMC) is a quantitative method of policy evaluation. It is oriented from *Omnia Mobilis*, which holds that all entities in the world are in motion and interconnected, and that every variable is equally important. In other words, all variables should be taken into account in policy evaluation.

Currently, PMC index model has been applied for policy evaluation in many fields such as health care, economic and social governance. Its application has been preliminary standardized, comprising the following four steps:

(1) **Selecting indicators and identifying parameters**

The selection and identification of variables form the foundation for policy evaluation. We primarily established the PMC index model based on the classification of policy variables in prior literature and the characteristics of high-frequency terms in policy texts. Specifically, previous studies indicated that the indicators could be categorized into three types: general objective indicators, general subjective indicators and characteristic indicators [29]. General objective indicators denote metrics that can be widely applied across various policies. Constructed based on a robust literature review, these indicators exhibit high universal relevance. *Policy nature, policy time, policy level and policy participation* were categorized under this type. General subjective indicators also possess broad applicability. But their evaluation is inherently subjective and necessitates integration with the specific content of the policies. *Policy evaluation* falls into this category. Characteristic indicators are tailored to the domain of laboratory biosafety. Based on the proceeding content analysis, *policy object, policy content, policy field and policy instrument* were identified as characteristic variables. The final index system comprises 9 primary variables and 37 secondary variables, as detailed in Table 1.

(2) **Building the multi-input-output table**

The multi-input-output table serves as a data analysis framework designed to store a large amount of data and enable multi-dimensional assessment of individual variables [30]. This framework integrates multiple primary indicators, each further divided into multiple secondary indicators. Establishing a multi-input-output table is fundamental to variable calculation. In this study, model indicators were first defined. Equal weights were then assigned to all the secondary indicators. Finally, all secondary variables within the PMC model were standardized into a binary format: a value of "1" was assigned when the policy content aligned with a secondary variable, and "0" was assigned in cases of non-compliance.

(3) **Calculating of the PMC-Index**

After constructing the multi-input-output table, sub-indicators were calculated using formula 1 and formula 2. Main-indicator values were subsequently derived via formula 3 and the final PMC index scores were computed using formula 4. Policy classification was conducted in accordance with the established PMC index rating framework proposed by Estrada, which categorizes policies into four tiers: *perfect* (8–9 points), *excellent* (6–7.99), *acceptable* (4–5.99), *inadequate* (0 to 3.99).

$$X \sim N[0, \ 1] \qquad\qquad\qquad \text{formula 1}$$

$$X = \{XR : \ [0 \sim 1]\} \qquad\qquad \text{formula 2}$$

$$Xi(\sum\nolimits_{j}^{n} \frac{X_{ij}}{T(X_{ij})}), \ i = 1,2,3,4,5,6,7,8,9,10\ldots\ldots m \qquad \text{formula 3}$$

$i$ = main indicator; $j$ = sub indicator; $n$ is the total number of the sub-indicator

$$PMC = \left\{ \sum\nolimits_{i=1}^{6} \frac{X_{1i}}{6} + \sum\nolimits_{j=1}^{3} \frac{X_{2j}}{3} + \sum\nolimits_{k=1}^{4} \frac{X_{3k}}{4} + \sum\nolimits_{l=1}^{2} \frac{X_{4l}}{2} + \sum\nolimits_{m=1}^{4} \frac{X_{5m}}{4} + \sum\nolimits_{n=1}^{4} \frac{X_{6n}}{4} + \sum\nolimits_{0=1}^{8} \frac{X_{7o}}{8} + \sum\nolimits_{p=1}^{3} \frac{X_{8p}}{3} + \sum\nolimits_{q=1}^{3} \frac{X_{9q}}{3} \right\}$$

(4) **Constructing of the PMC-Surface**

The calculated values for each dimension of the main indicators were transformed into a 3 × 3 matrix. Subsequently, a three-dimensional PMC surface was constructed to visualize the policies, thereby elucidating policy strengths and weaknesses and offering a more distinct representation of the consistency levels.

**Table 1. Evaluation Index for Laboratory Biosafety Policies.**

| Categories | First-Level Indicators | Second-Level Indicators |
|---|---|---|
| General-objective indicators | X1: Policy Nature | X1:1 Predictive |
| | | X1:2 Regulatory |
| | | X1:3 Advisory |
| | | X1:4 Descriptive |
| | | X1:5 Guiding |
| | | X1:6 Diagnostic |
| | X2: Policy Time | X2:1 Long-term |
| | | X2:2 Medium-term |
| | | X2:3 Short-term |
| | X3: Policy level | X3:1 State Council |
| | | X3:2 State Council departments |
| | | X3:3 State Council-affiliated institutions |
| | | X3:4 Others |
| | X4: Policy Participation | X4:1 Major department |
| | | X4:2 Other department |
| General-subjective indicators | X5: Policy evaluation | X5:1 Clear objectives |
| | | X5:2 Detailed content |
| | | X5:3 Sufficient evidence |
| | | X5:4 Scientific plan |
| Characteristic indicators | X6: Policy objective | X6:1 Provinces |
| | | X6:2 Department |
| | | X6:3 Affiliated institutions |
| | | X6:4 Others |
| | X7: Policy content | X7:1 Organization management |
| | | X7:2 Facilities, equipment, protective barriers |
| | | X7:3 Personnel management |
| | | X7:4 Management of bacterial (toxic) species and biological sample |
| | | X7:5 Experimental waste management |
| | | X7:6 Management of laboratory housekeeping, materials and identification |
| | | X7:7 Fire and security |
| | | X7:8 Other daily management |
| | X8: Policy field | X8:1 Society |
| | | X8:2 Environment |
| | | X8:3 Technology |
| | X9: Policy instrument | X9:1 Supply-side |
| | | X9:2 Environment-side |
| | | X9:3 Demand-side |

## Results

### Basic information of laboratory biosafety policies

**1. Annual and accumulated numbers of Laboratory biosafety policies**. After exclusion, 137 laboratory biosafety policies were included in the analysis. The annual number and cumulative distribution of these policies were presented

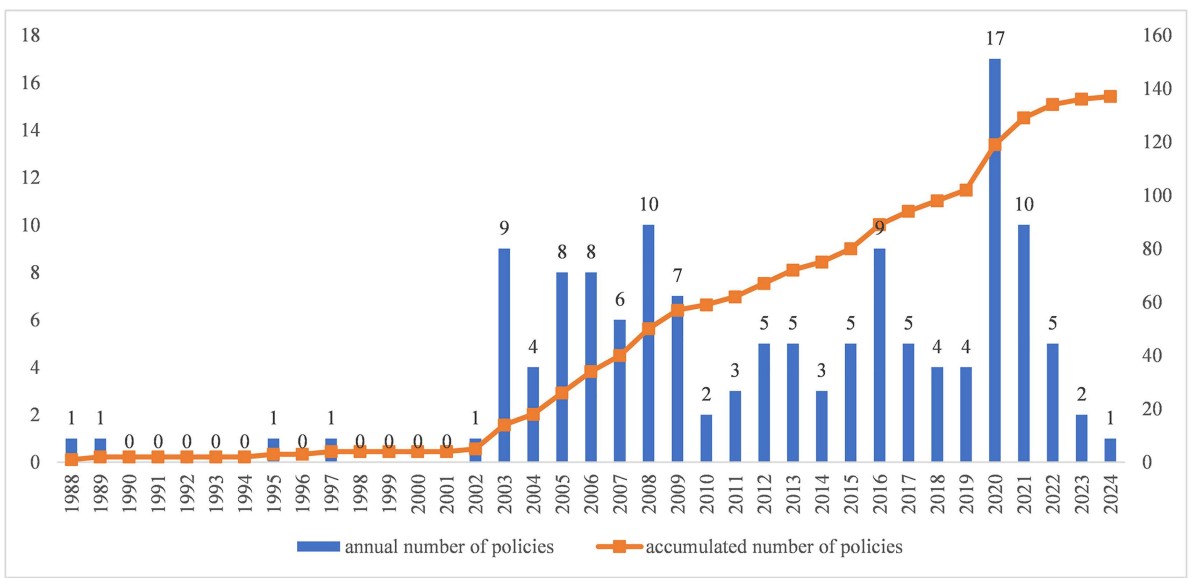

**Fig 1.  Annual and Accumulated Numbers of Laboratory Biosafety Policy from 1988 to 2024.**

in Fig 1. The earliest laboratory biosafety policy in China was promulgated in 1988 by the State Council, focusing on enhancing the laboratory animals management with provisions covering all stages from breeding to application. Then, scattered policies related to laboratory biosafety were intermittently issued from 1988 to 2002. Therefore, this period could be considered as the initial stage. A pronounced increase in policy issuance occurred from 2003 to 2019, termed the developmental phase in this analysis. Notably, the State Council issued the *Regulation on the Bio-safety Management of Patgogenic Microbe Labs* in 2004, which established standardized frameworks for laboratory biosafety management. The third phase, commencing in 2020, represented a transformative period in China's biosafety governance. The enactment of the *Biosafety Law* in 2020 introduced legally binding requirements, elevating laboratory biosafety management to a national strategic priority.

 **2. Promulgation departments of laboratory biosafety policies.** As shown in Table 2, a total of 24 departments contributed to the promulgation of laboratory biosafety policies. The National Health Commission emerged as the most active promulgating authority, responsible for 50 policies (42 issued independently and 8 as joint publications). This agency was primarily responsible for biosafety supervision of laboratories and experimental activities related to human health, encompassing laboratory biosafety management, public health emergency response, and the formulation of relevant laws and regulations. The Ministry of Agriculture and Rural Affairs ranked second, with 36 contributions, including 33 independent and 3 joint documents. Its mandate focused on the biosafety supervision of animal-related laboratories and experimental activities, covering laboratory biosafety in animal sector and animal diseases prevention and control.

 **3. Hierarchy analysis of policies.** As outlined in Table 3, China's laboratory biosafety management framework was comprised of three tiers: laws, administrative regulations and departmental regulations. There were four laws. For example, t*he Biosafety Law of the People's Republic of China* emphasized the need for the management of biosafety in pathogenic microorganism laboratories. Similarly, *the Law of the People's Republic of China on the Prevention and Control of Infectious Diseases* required strict supervision of laboratories handling pathogenic microorganism and biological samples, emphasizing standardized management protocols. Regarding the administrative regulations, there were a total of 14 items, including rules and plans. For instance, *the Biosafety Management Regulations for Pathogenic Microorganism Labs* specified measures for pathogen containment, laboratory operation, and infection control protocols. *The Notice of the*

**Table 2. Promulgation Departments of Laboratory Biosafety Policies.**

| Promulgation Department | Independent | Joint | Total |
|---|---|---|---|
| National Health Commission | 42 | 8 | 50 |
| Ministry of Agriculture and Rural Affairs | 33 | 3 | 36 |
| The State Council | 14 | 0 | 14 |
| The Joint Prevention and Control Mechanism of the State Council | 12 | 0 | 12 |
| Ministry of Science and Technology | 6 | 5 | 11 |
| Ministry of Ecology and Environment | 4 | 2 | 6 |
| State Administration of Traditional Chinese Medicine | 3 | 1 | 4 |
| Member of the Standing Committee of the National People's Congress | 4 | 0 | 4 |
| National Medical Products Administration | 1 | 2 | 3 |
| National Development and Reform Commission | 1 | 2 | 3 |
| State General Administration of the People's Republic of China for Quality Supervision and Inspection and Quarantine | 0 | 3 | 3 |
| Ministry of National Defense of the People's Republic of China | 0 | 2 | 2 |
| Ministry of Education | 0 | 2 | 2 |
| Standardization Administration of the People's Republic of China | 0 | 2 | 2 |
| China National Accreditation Service for Conformity Assessment | 2 | 0 | 2 |
| Ministry of Housing and Urban-Rural Development | 0 | 1 | 1 |
| Ministry of Finance | 0 | 1 | 1 |
| National Forestry and Grassland Administration | 0 | 1 | 1 |
| General Administration of Customs | 0 | 1 | 1 |
| Chinese Academy of Science | 0 | 1 | 1 |
| National Healthcare Security Administration | 0 | 1 | 1 |
| Civil Aviation Administration of China | 1 | 0 | 1 |
| Ministry of Public Security | 1 | 0 | 1 |

**Table 3. Hierarchy Analysis of Laboratory Biosafety Policy.**

| Level | number |
|---|---|
| Law | 4 |
| Administrative regulations | 14 |
| Departmental regulations | 119 |

*General Office of the State Council on Printing and Distributing the '14th Five-Year Plan' for National Health Development* further emphasized that enhanced laboratory biosafety oversight, proposing evaluation for high-level pathogen microbiology laboratories, and improved preservation system for culturable materials such as pathogenic microorganism strains (viruses) and experimental cells. Additionally, *Notice of the State Council on Printing and Distributing the Development Plan for the Bioindustry* advocated for strengthened biosafety system and regulatory frameworks. Departmental regulations constituted the largest proportion, with a total of 119, including measures, notices, opinions and guidance issued by relevant ministries and commissions. These documents aligned with their mandates, ensuring sector-specific biosafety implementation.

## Content analysis

**1. Analysis of Word Frequency**. Using ROST CM6 for word frequency analysis, redundant words like "about", "do well", "should" were identified and filtering out. The final top 30 most frequent words were presented in Table 4. Unsurprisingly,

**Table 4. High-frequency Words in Laboratory Biosafety Policy.**

| Keywords | Frequency | Keywords | Frequency |
|---|---|---|---|
| laboratory | 3588 | laboratory biosafety | 1263 |
| pathogenic microorganisms | 997 | management | 877 |
| animal | 858 | personnel | 745 |
| security | 694 | protection | 641 |
| medical | 484 | waste | 472 |
| sample | 457 | highly pathogenic | 451 |
| detection | 445 | experimental activity | 433 |
| unit | 420 | operation | 383 |
| disinfection | 372 | system | 362 |
| measure | 324 | preservation | 317 |
| environment | 294 | control | 290 |
| facility | 276 | device | 272 |
| experiment | 264 | material | 263 |
| transportation | 261 | high level | 254 |
| supervision | 245 | enhance | 244 |

given the policy focus, "laboratory" (3588 times), "biosafety" (1263 times), "pathogenic microorganisms" (997 times), and "management" (877 times) dominated. Subsequent analysis of policy keywords revealed that China's laboratory biosafety policy primarily addressed diverse management components. High-frequency terms such as "animals", "personnel", "waste", "samples", "units", "environment", and "materials" reflected a focus on core management elements. Additionally, terms like "protection," "sterilization," "facilities," and "devices" underscored the critical requirement for laboratories to maintain appropriate levels of infrastructure, safety barriers, and operational safeguards. This emphasis is essential for protecting laboratory personnel, ensuring experimental validity, and preventing environmental contamination. Furthermore, the frequent appearance of "testing", "experimental activities", "storage" and "transportation" highlighted the policy emphasis on standardizing laboratory procedures and operational techniques, given the complexity of tasks involved in biological research and handling.

In summary, these policies aimed to ensure laboratory biosafety through a multi-dimensional approach, encompassing facilities and equipment protection barriers, operational techniques, and management systems.

**2. Analysis of the Semantic Network.** We further analyzed the co-occurrence of the keywords and generated a co-occurrence matrix. The specific results were depicted in Fig 2. The network identified "laboratory", "biosafety", "pathogenic microorganisms" and "management" as core nodes, tightly interconnected with "personnel", "animals", "safety", "protection", "samples", "high pathogenicity", "experimental activities", "high level". This structure indicated that laboratory biosafety policies primarily focused on personnel management, animal management, biological sample management, safety protection and management of highly pathogenic microorganisms. Additionally, the prominent linkage to "high-level pathogenic microbiology laboratories" underscored the policy emphasis on specialized oversight for high-containment facilities.

## PMC index evaluation

**1. Sample selection.** PMC index model was mainly used for evaluation of specialized policies. Considering typical, representative, diversity and comprehensiveness of samples, 11 special policies were selected for evaluation. The details are presented in Table 5.

**2. PMC Index Calculation and Surface Construction Analysis.** Using the established policy evaluation index system, values were assigned to corresponding indicators for each policy, followed by calculation of PMC index

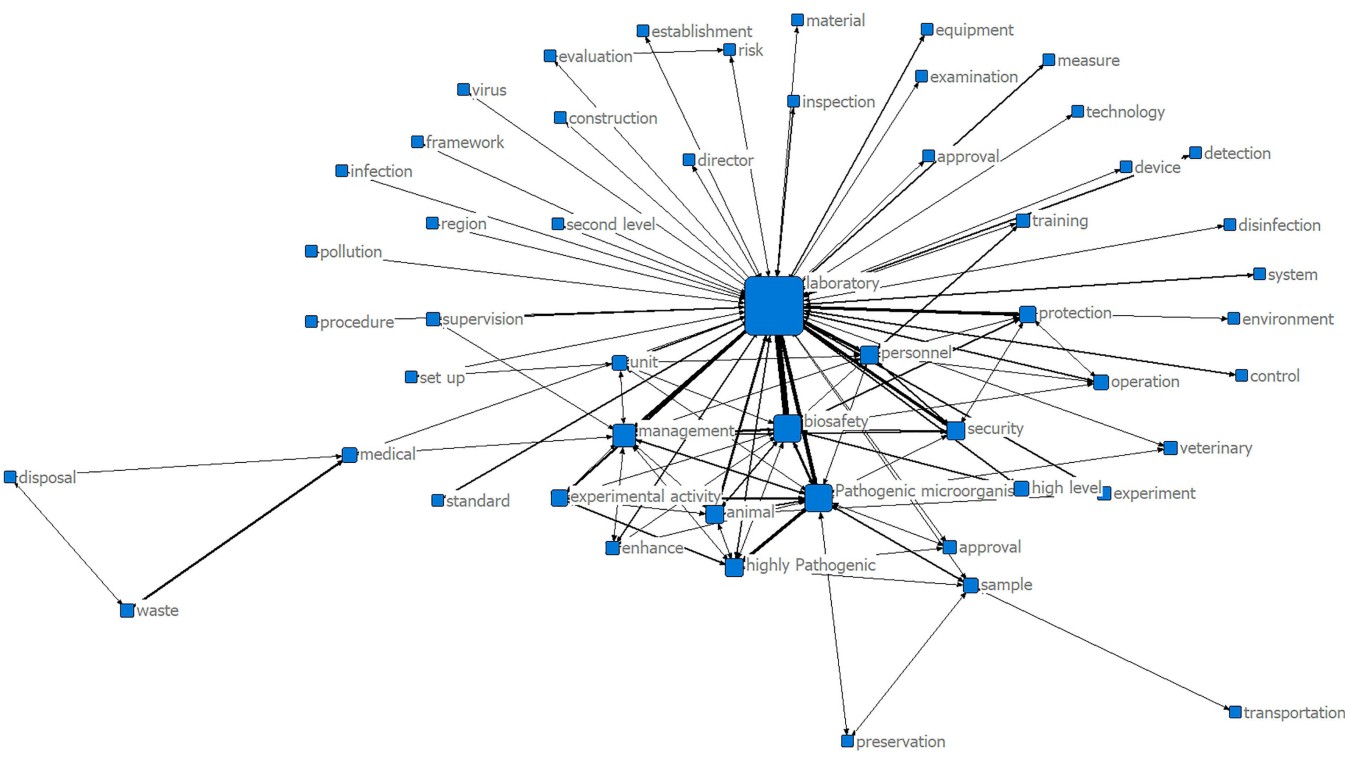

**Fig 2. Semantic Network of Laboratory Biosafety Policy.**

scores, and policies stratification according to the scores. The details were shown in Table 6. Additionally, surface plots were generated for each PMC index to visually characterize policy strengths and weakness. To illustrate this, three representative surface plots—corresponding to the highest, medium and lowest PMC index scores-- were presented in Fig 3.

The average PMC index of 11 representative policies was 5.05, spanning three tiers: Excellent, Acceptable, and Inadequate. Specifically, policies with a PMC index in the range of 6--7.99 were categorized as "Excellent", which included P1 and P4. P1, the *Regulation on the Biosafety Management of Pathogenic Microorganism Labs* issued by the State Council, boasted the highest index at 7.58 among all studied policies. The corresponding PMC surface plot (Fig 3a) revealed that this policy excelled across multiple dimensions, attaining perfect scores in 6 out of 9 indicators. Policies with a PMC index ranging from 4 to 5.99 were deemed as "Acceptable". The majority of policies (n = 8: P2, P3, P5, P6, P7, P9, P10, and P11) fell into the this tier. As an example, P7, *the Notice on Strengthening the Supervision and Management of Laboratory Biosafety in the Normalized Prevention and Control of COVID-19* issued by the National Health Commission, ranked sixth among all evaluated policies. Its PMC surface plot showed fewer protruding areas and more depressions, signifying numerous variables requiring enhancement(Fig 3b). Inadequate policies are those with a PMC index below 4. P8 was of this type. It is a notice on improving the supervision of biosafety laboratories in animal pathogenic microorganism, issued by the Ministry of Agriculture and Rural Affairs. The surface plot exhibited pronounced concavities, indicating several indicators were at extremely low level and necessitated improvement(Fig 3c).

Regarding specific indicators, the top-performing ones were policy instrument, policy evaluation, and policy participation. Conversely, the least-performing indicators were policy level, policy timeliness, and policy content.

**Table 5. 11 Laboratory Biosafety Policies Selected.**

| NO. | Policy Document | Issuing Department | Issuing Date |
|---|---|---|---|
| P1 | Regulation on the Bio-safety Management of Pathogenic Microbe Labs | State Council | Nov 12th, 2004 |
| P2 | The Planning for the Construction of High-level Bio-safety Laboratory Systems | National Development and Reform Commission, Ministry of Science and Technology | Nov 30th, 2016 |
| P3 | Management and Approval Measures for Biosafety Laboratories of Highly Pathogenic Animal Pathogens | Ministry of Agriculture and Rural Affairs | May 20th, 2005 |
| P4 | Notice on Strengthening Management of Animal Pathogenic Microorganism Laboratories biosafety | Ministry of Agriculture and Rural Affairs, Ministry of Education, Ministry of Science and Technology, National Health Commission, General Administration of Customs, National Forestry and Grassland Administration, Chinese Academy of Sciences | Feb 9th, 2020 |
| P5 | Notice on Enhancing Biosafety Management in COVID-19 Virus Laboratories | National Health Commission | Jan 17th, 2023 |
| P6 | Notice on the Issuance of the 'Management Measures for Large-scale COVID-19 Nucleic Acid Testing Laboratories (Trial) | The Joint Prevention and Control Mechanism of the State Council | Feb 17th, 2021 |
| P7 | Notice on Strengthening the Supervision and Management of Laboratory Biosafety in the Normalized Prevention and Control of COVID-19 | National Health Commission | July 6th, 2020 |
| P8 | Notice on Improving the Supervision of Biosafety Laboratories in Animal Pathogenic Microorganism | Ministry of Agriculture and Rural Affairs | March 8th, 2016 |
| P9 | Notice on Improving the Management of Biosafety Laboratories in Animal Pathogenic Microorganism | Ministry of Agriculture and Rural Affairs | May 11th, 2021 |
| P10 | Veterinary Laboratory Biosafety Guidelines | Ministry of Agriculture and Rural Affairs | Oct 15th, 2003 |
| P11 | Notice on Improving the biosafety management of pathogenic microorganism laboratories | National Health Commission | Sep 5th, 2006 |

## Discussion and conclusions

The outbreaks of SARS and COVID-19 prompted China to recognize the importance of laboratory biosafety management, leading to issuance of a series of policies aimed at enhancing regulatory frameworks management. In recent years, with the rapid advancement of biotechnology, traditional biosafety issues and new biosafety risks have overlapped, thereby exacerbating the complexity of laboratory landscape. Against this condition, it is of great significance to explore laboratory biosafety management from the policy-oriented perspective.

Text mining and PMC index model were used for textual analysis and quantitative evaluation of 137 national laboratory biosafety management policies. It indicated that a significant increase in policies issuance following the 2003 SARS outbreak, culminating in the establishment of a multi-tiered governance framework comprising laws, administrative regulations, and departmental regulations. These policies demonstrated broad coverage of critical domains, including equipment and facilities protection barriers, operational technical standards, and management systems. As for the policy evaluation, the average PMC index score of 11 policies was 5.05, falling within the acceptable range. This suggested the overall design of China's laboratory biosafety management policies was relatively sound. However, several challenges remain, primarily due to the delayed initiation of systematic laboratory biosafety management in the country.

Firstly, notification was the predominant format among the issued policies. This may be attributed to the flexibility inherent in the notification-based policy-making approach [31]. However, this reliance on the notifications may undermine policy continuity. Furthermore, the proliferation of such documents led to substantive content overlap, hindering the development

**Table 6. PMC index of 11 Laboratory Biosafety Policies.**

| Policy | X1 | X2 | X3 | X4 | X5 | X6 | X7 | X8 | X9 | PMC Index | Rating |
|--------|------|------|------|------|------|------|------|------|------|-----------|------------|
| P1 | 0.67 | 0.67 | 0.25 | 1.00 | 1.00 | 1.00 | 1.00 | 1.00 | 1.00 | 7.58 | Excellent |
| P2 | 0.67 | 0.33 | 0.25 | 1.00 | 1.00 | 0.50 | 0.38 | 0.67 | 1.00 | 5.79 | Acceptable |
| P3 | 0.33 | 0.33 | 0.25 | 0.50 | 0.75 | 0.50 | 0.38 | 0.67 | 0.67 | 4.38 | Acceptable |
| P4 | 0.50 | 0.33 | 0.75 | 0.50 | 0.75 | 0.75 | 0.50 | 1.00 | 1.00 | 6.08 | Excellent |
| P5 | 0.50 | 0.33 | 0.25 | 1.00 | 0.75 | 0.25 | 0.25 | 0.33 | 0.67 | 4.33 | Acceptable |
| P6 | 0.67 | 0.33 | 0.25 | 0.50 | 0.50 | 0.25 | 0.50 | 0.33 | 1.00 | 4.33 | Acceptable |
| P7 | 0.50 | 0.33 | 0.25 | 0.50 | 0.75 | 0.25 | 0.25 | 1.00 | 0.67 | 4.50 | Acceptable |
| P8 | 0.50 | 0.33 | 0.25 | 0.50 | 0.75 | 0.25 | 0.38 | 0.33 | 0.67 | 3.96 | Inadequate |
| P9 | 0.50 | 0.33 | 0.25 | 0.50 | 0.75 | 0.75 | 0.38 | 0.33 | 0.67 | 4.46 | Acceptable |
| P10 | 0.50 | 0.33 | 0.25 | 0.50 | 0.75 | 0.25 | 0.63 | 1.00 | 0.67 | 4.88 | Acceptable |
| P11 | 0.67 | 0.33 | 0.25 | 1.00 | 0.75 | 0.50 | 0.38 | 0.33 | 1.00 | 5.21 | Acceptable |
| Average | 0.55 | 0.36 | 0.30 | 0.68 | 0.77 | 0.48 | 0.45 | 0.64 | 0.82 | 5.05 | |

of a coherent regulatory framework. For instance, the Ministry of Agriculture and Rural Affairs released similar policy documents in 2016 and 2021, underscoring systemic redundancy.

Secondly, the study uncovered a lack of synergy in laboratory biosafety policies. Specifically, approximately 90% of policies were issued solely. Additionally, only two policies were jointly issued by two primary departments responsible for laboratory biosafety management, namely the National Health Commission and the Ministry of Agriculture and Rural Affairs. Even when addressing shared objectives related to strengthening laboratory biosafety management, these two agencies chose to issue separate documents rather than collaborate closely. This fragmented approach would result in inconsistent policy frameworks and insufficient systemic integration, ultimately impeding operational efficacy.

Thirdly, in accordance with the findings of previous studies, our analysis demonstrated that existing laboratory biosafety policies in China primarily centered on high-level laboratories, while overlooking low-level ones [32]. Lower-level biosafety laboratories, which accounted for the majority in China, encountered management challenges including a lack of targeted provisions, limited practicality, and implementation hurdles. For instance, many personnel were unclear about the difference between their designated responsibilities within biological laboratories and the corresponding duties of the host institution. Future policy initiatives should prioritize addressing these concerns.

Finally, PMC index analysis identified X2(policy time), X3(policy level), X6(policy objective) and X7(policy content) as weak indicators across all evaluated policies. For X2, current laboratory biosafety policies lacked provision for task decomposition timelines, which would undermine implementation scheduling. The low score for X3 aligned with the previous findings regarding the dominance of a single-department policy issuances. For X6, policies predominantly targeted administrative departments, with insufficient guidance for specific laboratories and laboratory establishment units. The suboptimal performance in X7 may stem from the multi-dimensional nature of laboratory biosafety management, which inherently requires cross-disciplinary coordination but was often addressed in fragmented policy language.

To foster the advancement of the laboratory management policies, the following recommendations are proposed:

1. **Strengthen communication among different departments.** A clear coordination mechanism and a standardized information-sharing platform should be established for laboratory biosafety. Regular cross-institutional joint meetings can facilitate communication and interaction among departments, ensuring timely transmission and exchange of relevant information. This approach enables effective management and prevents the issuance of duplicate policies across different departments.

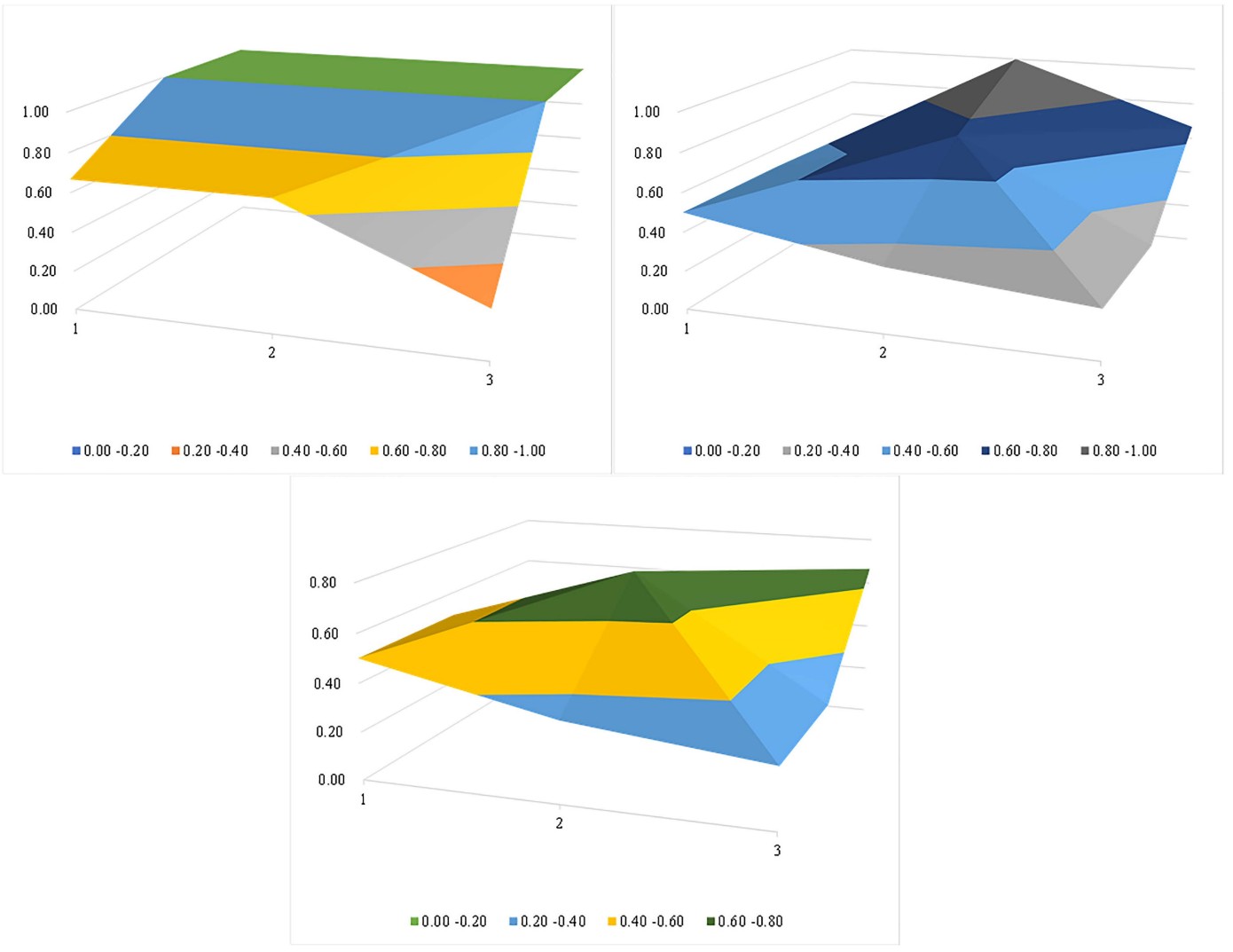

**Fig 3. PMC-Surface of representative policy. (a)** The PMC-Surface of P1. **(b)** The PMC-Surface of P7. **(c)** The PMC-Surface of P8.

2. **Optimize the policy formulation process.** Before drafting new policies, a comprehensive review of existing policies should be conducted, accompanied by expert discussions, and evidence-based investigations to ensure systemic coherence, logical consistency, and non-redundancy from a holistic perspective. In addition, when issuing policies, efforts should be made to minimize reliance on notifications as the primary format. Converting notifications into formal documents would enhance their systematicity, stability, sustainability, and enforceability.

3. **Enhance supervision of biosafety level 1 and 2 (BSL-1/2) laboratories.** A laboratory registration system has been established at the national level for biosafety level 2 laboratories; however, during implantation, registration format, procedures and validity period varies across provinces and cities, and the review process of the management part is also inconsistent [33]. Therefore, there is an urgent need to convene expert panels to develop standardized and implementable registration and management criteria to ensure consistency in the management of low-level biosafety laboratories. Additionally, it is recommended to enhance regulatory measures by formulating corresponding policies—for

example, encouraging the adoption of digital technologies in laboratory biosafety management and establishing a unified digital information management system to enable laboratory monitoring, early warning, and post-incident response.

4. **Harnessing the power of industry associations.** Under the national philosophy of multi-stakeholder collaboration, it is crucial to strategically engage industry forces, particularly by actively guiding industry associations to deeply participate in the development of biosafety quality control standards. This involvement serves to standardize the implementation of laboratory biosafety management protocols. Additionally, regular expert-led inspections of laboratory biosafety can be conducted to monitor and facilitate policy adherence.

This study has two main limitations: (1) The analysis mainly focused on the national-level polies, with limited examination of subnational policies. (2) The policy indicator system was developed based on the existing literature and textual analysis. In the future, continuous refinement of the indicator system is necessary to provide more scientifically robust, logically consistent, and comprehensive recommendations for policy enhancement.

## Supporting information

**S1 Appendix. 137 original policy documents.**
(DOCX)

**S2 Appendix. List of policy names.**
(DOCX)

## Author contributions

**Conceptualization:** Xueting Qiu, Fei Zhu.

**Data curation:** Cheng Xiang, Qing Zhu.

**Formal analysis:** Cheng Xiang.

**Methodology:** Cheng Xiang.

**Project administration:** Qing Zhu.

**Software:** Lixiang Xie.

**Visualization:** Jinqing Wen, Rongrong Hao.

**Writing – original draft:** Cheng Xiang.

**Writing – review & editing:** Xueting Qiu, Fei Zhu.

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
