## [Decision Letter · Decision Letter 0]

12 May 2025

Dear Dr. Zhu,

Thank you for submitting your manuscript to PLOS ONE. After careful consideration, we feel that it has merit but does not fully meet PLOS ONE’s publication criteria as it currently stands. Therefore, we invite you to submit a revised version of the manuscript that addresses the points raised during the review process.

We look forward to receiving your revised manuscript.

Kind regards,

Sanaullah Sajid, M.Phil/PhD

Academic Editor

PLOS ONE

Journal Requirements:

3. Please remove your figures from within your manuscript file, leaving only the individual TIFF/EPS image files, uploaded separately. These will be automatically included in the reviewers’ PDF.

Reviewers' comments:

Reviewer's Responses to Questions

**Comments to the Author**

1. Is the manuscript technically sound, and do the data support the conclusions?

Reviewer #1: Yes

2. Has the statistical analysis been performed appropriately and rigorously?

Reviewer #1: Yes

3. Have the authors made all data underlying the findings in their manuscript fully available?

Reviewer #1: Yes

4. Is the manuscript presented in an intelligible fashion and written in standard English?

Reviewer #1: No

Reviewer #1: Biosafety management system is not only worth studying in theory, but also in practice. Therefore, the topic of this paper deserves full affirmation. This paper is generally good, from the research method, data application and research content, are commendable. In terms of language and text, it is also necessary to optimize, and it is best to find professional institutions or people for native language modification. In the summary, we should briefly explain the countermeasures and suggestions. The countermeasures and suggestions in the last part of the article can be supplemented with corresponding discussions, like 4 and 5 in a simple sentence, which is a little irresponsible.

**Do you want your identity to be public for this peer review?** For information about this choice, including consent withdrawal, please see our Privacy Policy

Reviewer #1: **Yes: ** liu bangfan

---

## [Author Response · Author response to Decision Letter 1]

14 Jun 2025

Thank you very much for your suggestion. As for language, we tried our best to improve the manuscript and made some changes to the manuscript. These changes will not influence the content and framework of the paper. And here we did not list the changes but marked in red and blue in the revised paper.

Regarding the suggestions, we have fully considered the reviewers' comments and refined the originally proposed suggestions. In terms of strengthening communication among departments, the original suggestion was to accelerate the formation of a biosafety coordination mechanism, improve the information-sharing mechanism among departments, and establish an efficient working mechanism. Considering the feasibility of the suggestions, we have supplemented the content of regularly convening cross-departmental joint meetings in the revised version to enhance the feasibility of the suggestions.

In terms of the policy formulation process, we have mainly made revisions to the language expression without changing the content.

Regarding t Enhance supervision of biosafety level 1 and 2 (BSL-1/2) laboratories., considering that the filing system is an important policy for the management of Level 1 and 2 laboratories and that the current main issue lies in the differences in procedures and effectiveness across provinces and cities, we have adjusted the sequence of the two originally proposed suggestions.

In terms of leveraging the power of industry associations, the original version only briefly mentioned the rational use of non-governmental forces without clarifying how to bring their roles into play. The revised version has supplemented this content.

Regarding the original suggestion to set phased goals for long-term policies, considering that this involves issues related to policy-making procedures, we have deleted this item. The final version was as follows.

1. Strengthen communication among different departments. A clear coordination mechanism and a standardized information-sharing platform should be established for laboratory biosafety. Regular cross-institutional joint meetings can facilitate communication and interaction among departments, ensuring timely transmission and exchange of relevant information. This approach enables effective management and prevents the issuance of duplicate policies across different departments.

2. Optimize the policy formulation process. Before drafting new policies, a comprehensive review of existing policies should be conducted, accompanied by expert discussions, and evidence-based investigations to ensure systemic coherence, logical consistency, and non-redundancy from a holistic perspective. In addition, when issuing policies, efforts should be made to minimize reliance on notifications as the primary format. Converting select notifications into formal documents would enhance their systematicity, stability, sustainability, and enforceability.

3. Enhance supervision of biosafety level 1 and 2 (BSL-1/2) laboratories. A laboratory registration system has been established at the national level for biosafety level 2 laboratories; however, during implantation, registration format, procedures and validity period varies across provinces and cities, and the review process of the management part is also inconsistent[33]. Therefore, there is an urgent need to convene expert panels to develop standardized and implementable registration and management criteria to ensure consistency in the management of low-level biosafety laboratories. Additionally, it is recommended to enhance regulatory measures by formulating corresponding policies—for example, encouraging the adoption of digital technologies in laboratory biosafety management and establishing a unified digital information management system to enable laboratory monitoring, early warning, and post-incident response.

4. Harnessing the power of industry associations. Under the national philosophy of multi-stakeholder collaboration, it is crucial to strategically engage industry forces, particularly by actively guiding industry associations to deeply participate in the development of biosafety quality control standards. This involvement serves to standardize the implementation of laboratory biosafety management protocols. Additionally, regular expert-led inspections of laboratory biosafety can be conducted to monitor and facilitate policy adherence.

---

## [Decision Letter · Decision Letter 1]

9 Jul 2025

Mining and quantitative evaluation of the laboratory biosafety policy in China

PONE-D-24-55480R1

Dear Dr. Zhu,

We’re pleased to inform you that your manuscript has been judged scientifically suitable for publication and will be formally accepted for publication once it meets all outstanding technical requirements.

Kind regards,

Sanaullah Sajid, M.Phil/PhD

Academic Editor

PLOS ONE

Additional Editor Comments (optional):

Reviewers' comments:

Reviewer's Responses to Questions

**Comments to the Author**

Reviewer #2: (No Response)

2. Is the manuscript technically sound, and do the data support the conclusions?

Reviewer #2: Yes

3. Has the statistical analysis been performed appropriately and rigorously?

Reviewer #2: Yes

4. Have the authors made all data underlying the findings in their manuscript fully available?

Reviewer #2: Yes

5. Is the manuscript presented in an intelligible fashion and written in standard English?

Reviewer #2: No

Reviewer #2: The study is scientifically valid, methodologically rigorous, and ethically sound. It provides a valuable quantitative and qualitative evaluation of laboratory biosafety policies in China, addressing an important public health and policy issue. The analysis is well-structured and supported by robust data. However, minor language and editorial improvements are needed to enhance clarity and readability.

**Do you want your identity to be public for this peer review?** For information about this choice, including consent withdrawal, please see our Privacy Policy

Reviewer #2: No

---

## [Editor Report · Acceptance letter]

PONE-D-24-55480R1

PLOS ONE

Dear Dr. Zhu,

I'm pleased to inform you that your manuscript has been deemed suitable for publication in PLOS ONE. Congratulations! Your manuscript is now being handed over to our production team.

Kind regards,

on behalf of

Dr. Sanaullah Sajid

Academic Editor

PLOS ONE